# Redescription of *Bdella muscorum* Ewing, 1909 (Bdellidae: Bdellinae) from China with Its First Description of Ontogeny [note 1]

**DOI:** 10.3390/insects13121080

**Published:** 2022-11-23

**Authors:** Youfang Wu, Daochao Jin, Tianci Yi, Jianjun Guo

**Affiliations:** 1Institute of Entomology, Guizhou University, Guiyang 550025, China; 2Guizhou Provincial Key Laboratory for Agricultural Pest Management of the Mountainous Region, Guiyang 550025, China; 3Scientific Observing and Experimental Station of Crop Pests in Guiyang, Ministry of Agriculture and Rural Affairs of the China, Guiyang 550025, China

**Keywords:** Acari, *Bdella*, development, taxonomy, predator, China

## Abstract

**Simple Summary:**

Many species of mites are described as adults; immature stages are far less understood than adults despite being key features for understanding mite classification and phylogeny. *Bdella muscorum* Ewing, 1909 is a type species of Bdellidae erected on adults; and its ontogeny has never been described before. Here; adult *B. muscorum* collected from China were redescribed and its ontogeny was described in detail for the first time which will help us to improve identification, understand mite classification and phylogeny, and to study variations of ontogeny.

**Abstract:**

*Bdella muscorum* Ewing, 1909 was redescribed and illustrated in detail, and its ontogeny was described and illustrated for the first time, including pro dorsal apodeme and chaetotaxy. Chaetotaxy changes in *Bdella* are mainly focused on ventral hypostomal setae (*vh*), setae on palpal basifemur, aggenital setae (*ag*), genital setae (*g*), anal setae (*ad*) and leg setae. Furthermore, an original key to the *Bdella* species from China was also provided.

## 1. Introduction

Bdellidae Dugès, 1834 are active predators of small arthropods that have been shown to have an effective biological control on spider mites and springtails [1,2,3,4,5]. At present, Bdellidae includes five subfamilies, 11 genera, and 286 currently valid species [6,7,8,9,10,11].

*Bdella* Latreille, 1795 currently comprises 55 species all over the world, eight of which are from China [6,11,12]. *Bdella muscorum* Ewing, 1909 is widely distributed in many countries, including China. In China, however, knowledge on adult morphology is still limited. Furthermore, the ontogeny of this species is yet unknown, with no morphological specific information on its immature stages. With the advancement of technology, it is possible to obtain more in-depth knowledge on the species.

In this study, *B. muscorum* was redescribed and illustrated based on Chinese specimens in their immature stages, which may assist in enhancing the identification of *B. muscorum*, and *Bdella*. The immature stages were described for the first time. Meanwhile, an original key to the known Chinese *Bdella* is presented.

## 2. Materials and Methods

Samples were collected from moss, rotten wood, and plants in several provinces of China including Sichuan, Gansu, Qinghai, Guangdong, Hubei, Heilongjiang, Zhejiang, Tibet Autonomous Region, Xinjiang Uygur Autonomous Region, and Inner Mongolia autonomous region. Specimens were extracted from the samples by using Berlese-Tullgren funnels (homemade) for 12–24 h, mounted on slides in Hoyer’s medium [13], and examined under the microscope Nikon Ni E (Nikon corporation, Tokyo, Japan). Figures were drawn with the aid of a drawing tube attached to the microscope. The body lengths of all specimens were measured as previously described [10]. All life stages of the species that we collected were in the same habitat and the same place. We know that they are the same species based on distinguishing characteristics of the species. The nomenclature of the immatures is the same as that applied to the adults.

Abbreviations [10]: Prodorsal setae: anterior trichobothria (*at*), posterior trichobothria (*pt*), lateral proterosomal setae (*lps*), median proterosomal setae (*mps*). Hysterosomal setae: internal humerals (*c*_1_), external humerals (*c*_2_), internal dorsals (*d*_1_), internal lumbals (*e*_1_), internal sacrals (*f*_1_), external sacrals (*f*_2_), internal clunals (*h*_1_), external clunals (*h*_2_). Anal region: postanals (*ps*), anal setae (*ad*); Genital region: aggenital setae (*ag*), genital setae (*g*). Ventral hypostomal setae (*vh*). Leg setae: simple tactile seta (*sts*) attenuate solenidion (*asl*), blunt-pointed rod-like solenidion (*bsl*), peg-like seta (*pe*), trichobothria (*T*). Palp setae: dorsal end seta (*DES*), and ventral end seta (*VES*).

## 3. Results

Family Bdellidae Dugès, 1834Subfamily Bdellinae Grandjean, 1938Genus *Bdella* Latreille, 1795*Bdella muscorum* Ewing, 1909

### 3.1. Diagnosis

Subcapitulum with six pairs of ventral setae; prodorsal striae transverse and sparsely broken between *pt*; pedipalpal tibiotarsus with six setae (including two long terminal setae); palp basifemur with 9–11 setae; prodorsal apodeme reticulated; genu II with duplex setae; genu IV without duplex setae.

### 3.2. Description

Adults, Female (n = 37) (Figure 1, Figure 2 and Figure 3)

Body length (including gnathosoma) 1230–1414, idiosoma 911–1068, body width 537–635.

Gnathosoma (Figure 1A–C). Gnathosoma length 295–350, width 98–122. Subcapitulum with longitudinal striae from buccal cone to *vh*_2_, transverse striae at the base, and six pairs of ventral setae (*vh*_1_–*vh*_6_) (Figure 1A). Chelicera length 285–337, with longitudinal striae broken at the base of chelicera, two setae, fixed and movable digits smooth (Figure 1B). Palp five-segmented (Figure 1C), total length 299–352, trochanter 15–17, basifemur 145–175, telofemur 30–39, genu 22–27, tibiotarsus 81–96, *VES* 185–201, *DES* 233–267; palp chaetotaxy: trochanter 0, basifemur 10–11 *sts*, telofemur 1 *sts*, genu 4 *sts*, tibiotarsus 3 *sts*, 1 *asl*, two long terminal setae (*VES*, *DES*) (Table 1).

Dorsum (Figure 2A). Prodorsum with transverse striae, prodorsal apodeme reticulated between *at* and *pt*, anterior region of prodorsum with elongated chamber. Two pairs of lateral eyes, the anterior one 23–25; the posterior one 20–24. Hysterosoma with broken or continuous striae; with three cupules (*ia*, *im* and *ip*) at the level of setae *d*_1_, *e*_1_ and between *f*_1_; all dorsal setae smooth. Measurements of dorsal setae as follows: *at* 154–201, *pt* 184–256, *lps* 51–68, *mps* 85–114, *c*_1_ 87–119, *c*_2_ 122–133, *d*_1_ 73–88, *e*_1_ 70–88, *f*_1_ 68–85, *f*_2_ 72–97, *h*_1_ 87–104, *h*_2_ 72–83.

Venter (Figure 2B). Broken striae longitudinal between coxae I–II and III–IV, and transverse between coxae II–III. Aggenital region with 11 pairs of setae (*ag*_1*–*11_); genital valves with eight pairs of setae (*g*_1*–*8_); three pairs of genital papillae; anal region with three pairs of postanals (*ps*_1*–*3_) and three pairs of anal setae (*ad*_1*–*3_) (Table 1).

Ovipositor (Figure 1D). With 12 dorsal and seven ventral setae.

Legs (Figure 3). Measurements of legs as follows: leg I 576–678, leg II 523–598, leg III 597–712, leg IV 719–846; setal formula of leg segments as follows: coxae I–IV 6-6-5-3 *sts*; trochanters I–IV 1-1-2-2 *sts*; basifemora I–IV 12(13)-9(8)-11(9)-5 *sts*; telofemora I–IV 9(10)-9(8)-5(6)-8 *sts*; genua I–IV 6 *sts*, 2 *asl*, 1 *dxs*-6 *sts*, 1 *dxs*-6 *sts*, 1 *dxs*-8 *sts*, 1 *asl*; tibiae I–IV 14 *sts*, 3 *asl*, 1 *dxs*, 1 *T*-11 *sts*, 2 *asl*, 1 *bsl*-12 *sts*, 1 *asl*-13 *sts*, 1 *T*; tarsi I–IV 27 *sts*, 2 *asl*, 2 *bsl*, 1 *pe*-23 *sts*, 2 *bsl*, 1 *pe*-27 *sts*, 1 *T*-23 *sts*, 1 *asl*, 1 *T* (Table 2).

Tritonymph (n = 36) (Figure 4, Figure 5 and Figure 6).

Body length (including gnathosoma) 995–1259, idiosoma 752–978, body width 491–657.

Gnathosoma (Figure 4). Gnathosoma length 238–290, width 103–189. Subcapitulum striae resembling female, with five pairs of *vh* (Figure 4A). Chelicera length 238–258, striae resembling female (Figure 4B). Palp similar to female (Figure 4C); total length 247–278, trochanter 12–15, basifemur 117–141, telofemur 23–28, genu 19–28, tibiotarsus 62–73, *VES* 155–179, *DES* 201–217; palp chaetotaxy: trochanter 0, basifemur 7 *sts*, telofemur 1 *sts*, genu 4 *sts*, tibiotarsus 3 *sts*, 1 *asl*, two long terminal setae (*VES*, *DES*) (Table 1).

Dorsum (Figure 5A). Striae resembling female; prodorsal apodeme simpler and weaker; Two pairs of lateral eyes, the anterior one 17–20, and the posterior one 15–19. Measurements of dorsal setae as follows: *at* 165–190, *pt* 168–246, *lps* 49–57, *mps* 84–104, *c*_1_ 91–98, *c*_2_ 115–127, *d*_1_ 74–87, *e*_1_ 70–80, *f*_1_ 73–82, *f*_2_ 63–82, *h*_1_ 89–106, *h*_2_ 71–92.

Venter (Figure 5B). Resembling female; nine pairs of *ag*; five pairs of *g*; three pairs of *ps* and three pairs of *ad* (Table 1).

Legs (Figure 6). Measurements of legs as follows: leg I 486–525, leg II 415–452, leg III 528–570, leg IV 602–644; setal formula of leg segments as follows: coxae I–IV 5-5-5-3 *sts*; trochanters I–IV 1-1-2-1 *sts*; basifemora I–IV 8-7-7-3 *sts*; telofemora I–IV 6-5-6-5 *sts*; genua I–IV 5 *sts*, 2 *asl*, 1 *dxs*-5 *sts*, 1 *dxs*-4 *sts*, 1 *dxs*-6 *sts*, 1 *a*sl; tibiae I–IV 8 *sts*, 2 *asl*, 1 *dxs*, 1 *T*-7 *sts*, 2 *asl*, 1 *bsl*-7 *sts*, 1 *asl*-8 *sts*, 1 *T*; tarsi I–IV 26 *sts*, 2 *asl*, 2 *bsl*, 1 *pe*-21 *sts*, 2 *bsl*, 1 *pe*-21 *sts*, 1 *T*-19 *sts*, 1 *asl*, 1 *T* (Table 2).

Deutonymph (n = 17) (Figure 7, Figure 8 and Figure 9).

Body length (including gnathosoma) 777–886, idiosoma 556–685, body width 359–441.

Gnathosoma (Figure 7). Gnathosoma length 200–228, width 74–112. Subcapitulum resembling tritonymph but with four pairs of *vh* (Figure 7A). Chelicera length 184–198, resembling tritonymph (Figure 7B). Palp similar to tritonymph (Figure 7C); total length 203–217, trochanter 10–12, basifemur 95–98, telofemur 22–28, genu 19–20, tibiotarsus 53–59, *VES* 106–133, *DES* 130–171; palp chaetotaxy: trochanter 0, basifemur 6 *sts*, telofemur 1 *sts*, genu 4 *sts*, tibiotarsus 3 *sts*, 1 *asl*, two long terminal setae (*VES*, *DES*) (Table 1).

Dorsum (Figure 8A). Resembling tritonymph; prodorsal apodeme simple without reticulation; two pairs of lateral eyes, the anterior one 15–17, and the posterior one 14–15. Measurements of dorsal setae as follows: *at* 113–114, *pt* 139–197, *lps* 36–44, *mps* 71–83, *c*_1_ 60–74, *c*_2_ 91–108, *d*_1_ 56–64, *e*_1_ 53–58, *f*_1_ 54–63, *f*_2_ 54–62, *h*_1_ 71–78, *h*_2_ 50–55.

Venter (Figure 8B). Resembling tritonymph; five or six pairs of *ag*; two pairs of *g*; three pairs of *ps* and three pairs of *ad* (Table 1).

Legs (Figure 9). Measurements of legs as follows: leg I 377–418, leg II 331–361, leg III 394–439, leg IV 429–476; setal formula of leg segments as follows: coxae I–IV 5/4-3-3/5-2 *sts*; trochanters I–IV 1-1-2-1 *sts*; basifemora I–IV 7-7-4-1 *sts*; telofemora I–IV 5-5-5-4 *sts*; genua I–IV 4 *sts*, 1 *asl*, 1 *dxs*-4 *sts*, 1 *dxs*-4 *sts*, 1 *dxs*-4 *sts*, 1 *asl*; tibiae I–IV 7 *sts*, 2 *asl*, 1 *dxs*, 1 T-5 *sts*, 2 *asl*, 1 *bsl*-5 *sts*, 1 *bsl*-7 *sts*, 1 *T*; tarsi I–IV 19 *sts*, 2 *asl*, 2 *bsl*, 1 *pe*-16 *sts*, 2 *bsl*, 1 *pe*-17 *sts*, 1 *T*-15 *sts*, 1 *asl*, 1 *T* (Table 2).

Protonymph (n = 7) (Figure 10, Figure 11 and Figure 12).

Body length (including gnathosoma) 528–602, idiosoma 385–442, body width 224–286.

Gnathosoma (Figure 10). Gnathosoma length 141–170, width 54–62. Subcapitulum resembling deutonymph but with three pairs of *vh* (Figure 10A). Chelicera length 141–163, resembling deutonymph (Figure 10B). Palp similar to deutonymph (Figure 10C); total length 142–175, trochanter 7–11, basifemur 67–77, telofemur 13–22, genu 16–24, tibiotarsus 42–55, *VES* 91–104, *DES* 126–133; palp chaetotaxy: trochanter 0, basifemur 3 *sts*, telofemur 1 *sts*, genu 4 *sts*, tibiotarsus 3 *sts*, 1 *asl*, two long terminal setae (*VES*, *DES*) (Table 1).

Dorsum (Figure 11A). Resembling deutonymph; prodorsal apodeme very faint; two pairs of lateral eyes, the anterior one 12–15, and the posterior one 12–13. Measurements of dorsal setae as follows: *at* 112–120, *pt* 142–153, *lps* 29–39, *mps* 53–61, *c*_1_ 46–60, *c*_2_ 84–86, *d*_1_ 40–50, *e*_1_ 43–51, *f*_1_ 46–61, *f*_2_ 49–53, *h*_1_ 62–65, *h*_2_ 38–47.

Venter (Figure 11B). Resembling deutonymph; one pair of *ag*; one pair of *g*; three pairs of *ps* and two pairs of *ad* (Table 1).

Legs (Figure 12). Measurements of legs as follows: leg I 284–307, leg II 238–276, leg III 294–334, leg IV 276–308; setal formula of leg segments as follows: coxae I–IV 4-2-4-0 *sts*; trochanters I–IV 1-1-2-0 *sts*; basifemora I–IV 2-2-2-0 *sts*; telofemora I–IV 5-5-4-0 *sts*; genua I–IV 4 *sts*, 1 *asl*, 1 *dxs*-4 *sts*, 1 *dxs*-4 *sts*, 1 *dxs*-0; tibiae I–IV 4 *sts*, 1 *asl*, 1 *dxs*, 1 *T*-5 *sts*, 1 *asl*, 1 *bsl*-5 *sts*, 1 *bsl*-1 *sts*; tarsi I–IV 17 *sts*, 2 *asl*, 1 *bsl*, 1 *pe*-15 *sts*-13 *sts*, 1 *T*-7 *st*s (Table 2).

Larva (n = 7) (Figure 13 and Figure 14).

Body length (including gnathosoma) 424–506, idiosoma 303–380, body width 196–260.

Gnathosoma (Figure 13). Gnathosoma length 110–134, width 50–57. Subcapitulum striae weak, with two pairs of *vh* (Figure 13A). Chelicera length 101–117, resembling protonymph (Figure 13B). Palp (Figure 13C) total length 125–139, trochanter 7–9, basifemur 50–58, telofemur 13–18, genu 14–16, tibiotarsus 33–41, *VES* 67–78, *DES* 86–95; palp chaetotaxy: trochanter 0, basifemur 2 *sts*, telofemur 1 *sts*, genu 4 *sts*, tibiotarsus 3 *sts*, 1 *asl*, two long terminal setae (*VES*, *DES*) (Table 1).

Dorsum (Figure 14A). Resembling protonymph; prodorsal apodeme almost invisible; two pairs of lateral eyes, the anterior one 8–12, and the posterior one 8–10. Measurements of dorsal setae as follows: *at* lost, *pt* lost, *lps* 27–31, *mps* 49–59, *c*_1_ 42–48, *c*_2_ 59–72, *d*_1_ 36–45, *e*_1_ 35–41, *f*_1_ 34–49, *f*_2_ 37–50, *h*_1_ 39–47, *h*_2_ 35–45.

Venter (Figure 14B). Without genital valves, genital and aggenital setae; three pairs of *ps*, without *ad* (Table 1).

Legs (Figure 13D–F). Measurements of legs as follows: leg I 206–249, leg II 193–225, leg III 219–283; setal formula of leg segments as follows: coxae I–III 2-1-2 *sts*; trochanters I–III 0-0-0 *sts*; femora I–III 6-6-5 *sts*; genua I–III 4 *sts*, 1 *asl*, 1 *dxs*-4 *sts*, 1 *dxs*-4 *sts*, 1 *dxs*; tibiae I–III 4 *sts*, 1 *asl*, 1 *dxs*, 1 T-5 *sts*, 1 *asl*, 1 *bsl*-5 *sts*, 1 *bsl*-1 *sts*; tarsi I–III 14 *sts*, 1 *asl*, 1 *bsl*, 1 *pe*-13 *sts*, 1 *bsl*, 1 *pe*-11 *sts*, 1 *T* (Table 2).

Male unknown.

### 3.3. Voucher Material

Eight female adults (slide NO. NMG-BD-I-20180701–20180708), eight tritonymphs (slide NO. NMG-BD-II-20180701–20180706, NMG-BD-II-20170701–20170702), five deutonymphs (slide NO. NMG-BD-III-20180701–20180705), four protonymphs (slide NO. NMG-BD-IV-20180701–20180704), and five larvae (slide NO. NMG-BD-V-20180701–20180705) were collected from moss in Hanma National Nature Reserve, Genhe City, Inner Mongolia Autonomous Region, China (N51°45′40.7772″, E122°29′59.8056″, H 1025 m; N51°19′41.65″, E121°28′33.88″, H 782 m; N51°45′42.6734″, E122°29′20.9321″, H 1025 m) by Yun Long and Maoyuan Yao in July 2018 and August 2017. Two female adults (slide NO. XZ-BD-I-20190701–20190702), one tritonymph (slide NO. XZ-BD-II-20190701), three deutonymphs (slide NO. XZ-BD-III-20190701–20190703), and one larva (slide NO. XZ-BD-V-20190701) were collected from moss in Bomi County, Nyingchi City, Tibet Autonomous Region, China (N29°31′55.35″, E96°33′10.44″, H 3712 m) by Jianxin Chen in July 2019. 11 female adults (slide NO. XJ-BD-I-20190701–20190711), 19 tritonymphs (slide NO. XJ-BD-II-20190701–20190719), four deutonymphs (slide NO. XJ-BD-III-20190701–20190704), one protonymph (slide NO. XJ-BD-IV-20190701), and one larva (XJ-BD-V-20190701) were collected from moss and rotten wood in Baihaba National Forest Park, Burqin County, Altay, Xinjiang Uygur Autonomous Region, China (N48°30′5.48″, E87°8′12.05″, H 1341 m; N48°42′22.59″, E87°2′0.92″, H 1377 m; N48°38′46.50″, E86°42′40.10″, H 1300 m) by Jianxin Chen in July 2019. Two female adults (slide NO. GS-BD-I-20180801–20180802) and one tritonymph (slide NO. GS-BD-II-20180801, GS-BD-II-20190701) were collected from moss in Longnan City and Lujiao Valley in Qilian Mountains, Gansu Province, China (N33°3′25″, E104°42′31″, H 1800 m; N38°12′9122″, E100°43′4932″, H 3150 m) by Guoru Ren and Qianfeng Zheng in August 2018 and July 2019. Two female adults (slide NO. SC-BD-I-20190501–20190502) were collected from fallen leaves and bark in Tangjiahe Nature Reserve, Qingchuan County, Sichuan Province, China (N34°34′3.3924″, E104°47′56.6296″, H 1220 m; N32°34′0.38688″, E104°48′8.478″, H 1114 m) by Yun Long in May 2019. Two female adults (slide NO. QH-BD-I-20200701–20200702) and two tritonymphs (slide NO. QH-BD-II-20200701–20200702) were collected from *Potentilla fruticose* in Sanjiangyuan National Reserve in Qinghai Province, China (N35°18′6.06″, E101°52′48.8460″, H 3190 m; N35°13′17.4072″, E101°57′9.5724″, H 3027 m) by Dongdong Li in July 2020. One female adult (slide NO. GD-BD-I-20190401) and one deutonymph (slide NO.GD-BD-III-20190401) were collected from fallen leaves in Nanling National Forest Park, Ruyuan Yao Autonomous County, Shaoguan City and Yunkai Mountain National Nature Reserve, Xinyi City, Guangdong Province, China (N24°56′55.17″, E112°59′29.16″, H 903 m; N22°17′20.96″, E111°13′21.84″, H 1563 m) by Jianxin Chen and Boyan Li in April 2019. Three female adults (slide NO. HB-BD-I-20180801–20180803) were collected from fallen leaves in Xingdoushan National Nature Reserve, Enshi Tujia and Miao Autonomous Prefecture, Hubei Province, China (N30°02′04″, E109°07′40″, H 1680 m) by Jianxin Chen and Xuesong Zhang in August 2018. Five female adults (slide NO. HLJ-BD-I-20190801–20190805), four tritonymphs (slide NO. HLJ-BD-II-20190801–20190804), four deutonymphs (slide NO. HLJ-BD-III-20190801–20190804) and two protonymphs (slide NO. HLJ-BD-IV-20190801–20190802) were collected from moss and fallen leaves in Wudalianchi City, Heilongjiang Province, China (N48°41′137″, E127°4′21″, H 283 m; N48°47′24″, E127°5′46″, H 366 m) by Min Ao in August 2019. One female adult (slide NO. ZJ-BD-I-20180701) was collected from moss in Tianmu Mountain National Nature Reserve, Hangzhou City, Zhejiang Province (N30°19′18.6348″, E119°26′38.2344″, H 387 m) by Maoyuan Yao and Yun Long in July 2018.

All voucher specimens are deposited in GUGC, the Institute of Entomology, Guizhou University, Guiyang, China [14].

## 4. Discussion

### 4.1. Species Identification

In the current study, the specimens collected from China are the same as *Bdella muscorum* based on the following features: (1) subcapitulum with six pairs of ventral setae (*vh*); (2) trichobothrium absent from tibiae II, present on tarsi IV; (3) palp chaetotaxy: 0-10(11)-1-4-6; (4) prodorsal striae between *at* and *pt* continuous to sparsely broken transverse striations; (5) prodorsal apodeme reticulated between *at* and *pt*, anterior region of prodorsum with elongated chamber; and (6) genua II with duplex setae, genua IV without duplex setae [15,16,17].

These specimens have a few differences as follows: (1) ovipositor with 19 setae in China specimens, but 16 setae in Iranian specimens and 18 setae in American specimens; (2) coxae IV with three setae in Chinese specimens and American specimens, but with four in Iranian specimens; (3) aggenital region with 11 pairs of setae in Chinese specimens, but with nine pairs of setae in Iranian specimens and 10 pairs of setae in American specimens. According to the description of *B. muscorum* from different countries, we found the numbers of leg setae are different in different areas, which is relatively inconstant [15,16,17]. Eghbalian & Khanjani [17] have also investigated the differences of *B. muscorum* striae from different countries and categorized these differences as regional traits.

### 4.2. Ontogeny of B. muscorum

Although immature stages play a significant role in mite classification and phylogeny, they are far less well-known than their adult forms [18]. All previous descriptions of *B. muscorum* have been adult-focused, without any description of the immature stages. Furthermore, chaetotaxy is an important feature that can be applied in species identification, classification, phylogenetics, ontogeny, and so on [11,19,20,21].

The current study examines the first report on the description of *B. muscorum* in its immature phases. The changes of ontogeny mainly focus prodorsal apodeme and chaetotaxy.

Our results show that prodorsal apodeme is almost invisible in larva but as the stages develop, prodorsal apodeme and its reticulation become clearer and clearer.

Chaetotaxy changes mainly focus on ventral hypostomal setae (*vh*), setae on palpal basifemur, aggenital setae (*ag*), genital setae (*g*), adanal setae (*ad*), and leg setae. *B. muscorum* follows the same change rule as other species of *Bdella* [11]: subcapitulum with two pairs of *vh* on larvae, three pairs on protonymphs, four on deutonymphs, five on tritonymphs and six on adults; palpal setae on basifemur increase with the stages, two setae on larvae, three setae on protonymphs, six on deutonymphs, seven on tritonymphs and 10 or 11 on adults; genital region undeveloped in larvae, no genital plates, *g* and *ag* in larva, one *g* and *ag* on protonymphs, then *g* and *ag* increase with the development of life stages; anal region no *ad* on larvae, two *ad* on protonymphs, three *ad* on deutonymphs, tritonymphs and adults; *ps* does not change among stage; three pairs of legs on larvae, four pairs of legs in subsequent stages, and the setae of leg increases with each life stage.

In this paper, the detailed description of *B. muscorum* will help us to improve its identification and understanding of ontogeny, and provide useful documentation and material for studying the ontogeny of mites.

### 4.3. Key to Adult Bdella of China

1. Prodorsal striae between *at* and *pt* longitudinal..……………….…….……….………2

    Prodorsal striae between *at* and *pt* transverse..…………………………………………5

2. Hysterosomal setae branched distally…………….*B. distincta* Baker & Balock, 1944

    Hysterosomal setae smooth distally………………………………………………………3

3. Palpal basifemur with six setae……………………………………*B. tropica* Atyeo, 1960

    Palpal basifemur with more at least 11 setae…………………………………………4

4. Palpal basifemur with 11 setae…………………………………*B. iconica* Berlese, 1923

    Palpal basifemur with 14 setae……………………………………*B. uchidai* Ehara, 1961

5. Palpal tibiotarsus with six setae…………………………………………………………6

    Palpal tibiotarsus with seven setae……………………………………………………7

6. Palpal basifemur with 9–11setae……………………………*B. muscorum* Ewing, 1909

    Palpal basifemur with 14–15 setae………………………………*B. xini* Wu & Guo, 2021

7. Prodorsal striae between *at* and *pt* sparsely broken trans verse…………………………………………………………*B. semiscutata* Thor, 1930

    Prodorsal striae between *at* and *pt* continuously trans verse……………………………………………………*B. longicornis* (Linnaeus, 1758)

## Figures and Tables

**Figure 1 insects-13-01080-f001:**
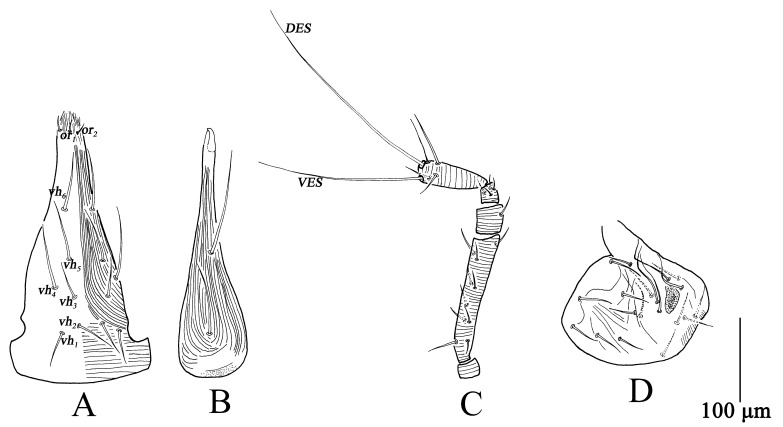
*Bdella muscorum*, female: **(A**) Subcapitulum; **(B**) Chelicera; **(C**) Palp; **(D**) Ovipositor.

**Figure 2 insects-13-01080-f002:**
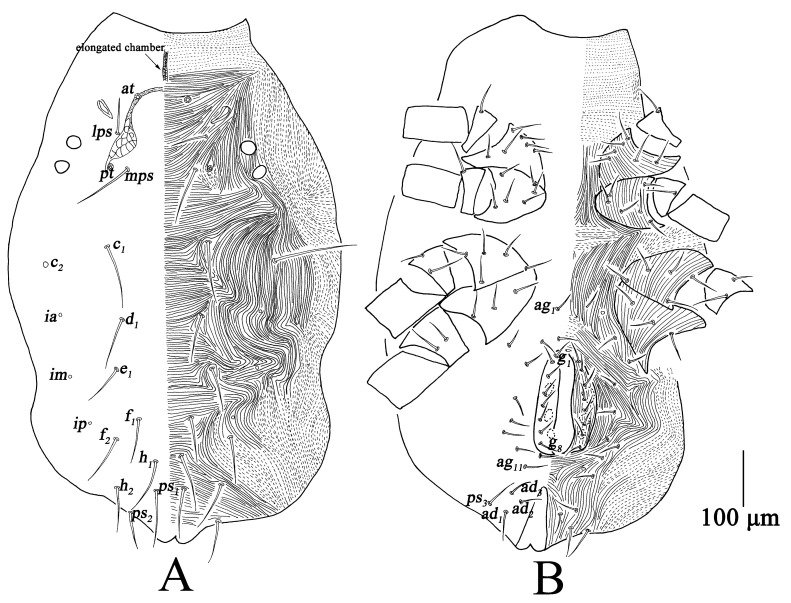
*Bdella muscorum*, female: (**A**) Dorsal view of idiosoma; (**B**) Ventral view of idiosoma.

**Figure 3 insects-13-01080-f003:**
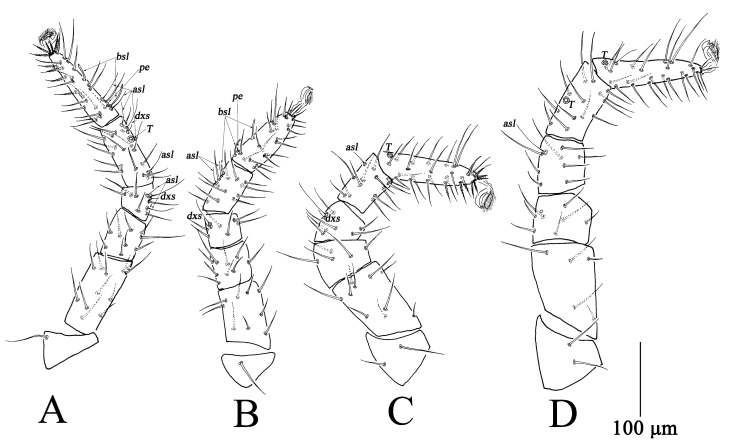
*Bdella muscorum*, female: (**A**) Leg I; (**B**) Leg II; (**C**) Leg III; (**D**) Leg IV.

**Figure 4 insects-13-01080-f004:**
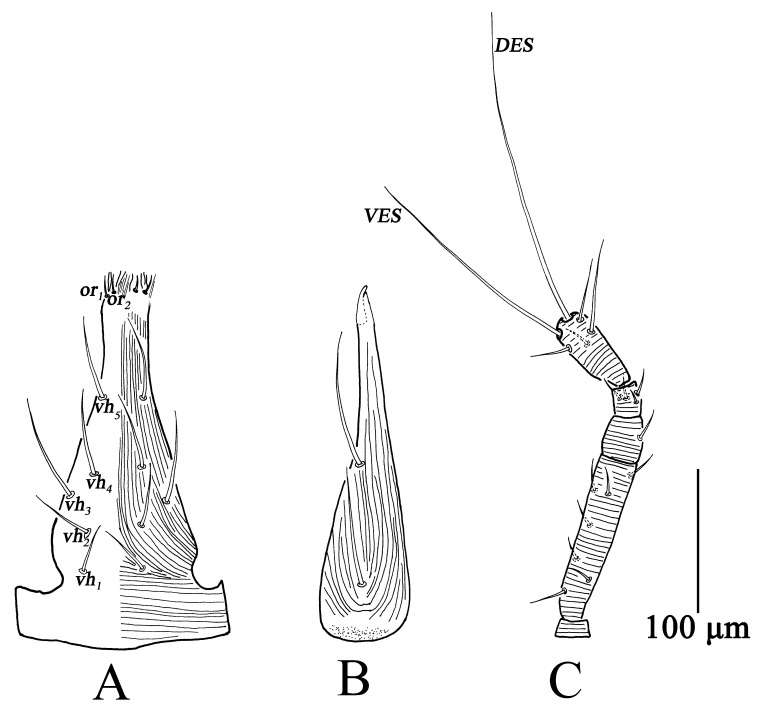
*Bdella muscorum*, tritonymph: (**A**) Subcapitulum; (**B**) Chelicera; (**C**) Palp.

**Figure 5 insects-13-01080-f005:**
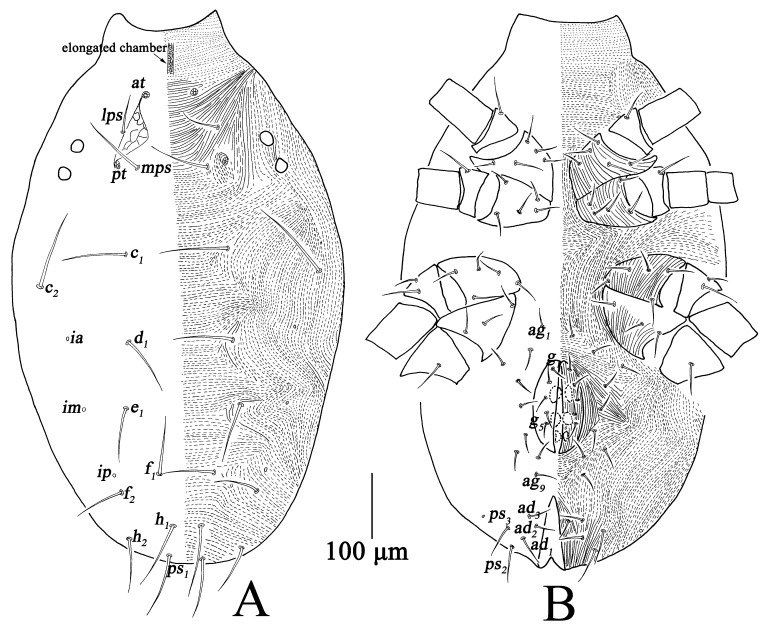
*Bdella muscorum*, tritonymph: (**A**) Dorsal view of idiosoma; (**B**) Ventral view of idiosoma.

**Figure 6 insects-13-01080-f006:**
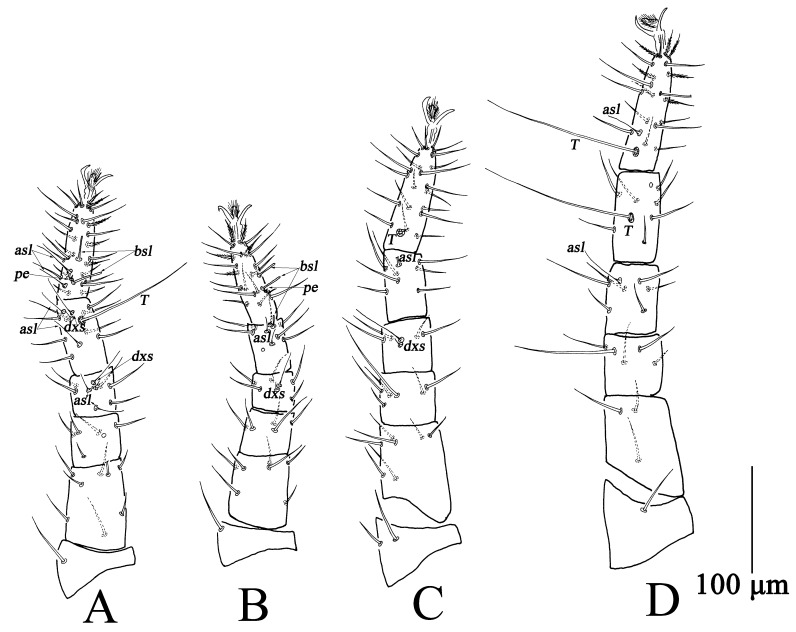
*Bdella muscorum*, tritonymph: (**A**) Leg I; (**B**) Leg II; (**C**) Leg III; (**D**) Leg IV.

**Figure 7 insects-13-01080-f007:**
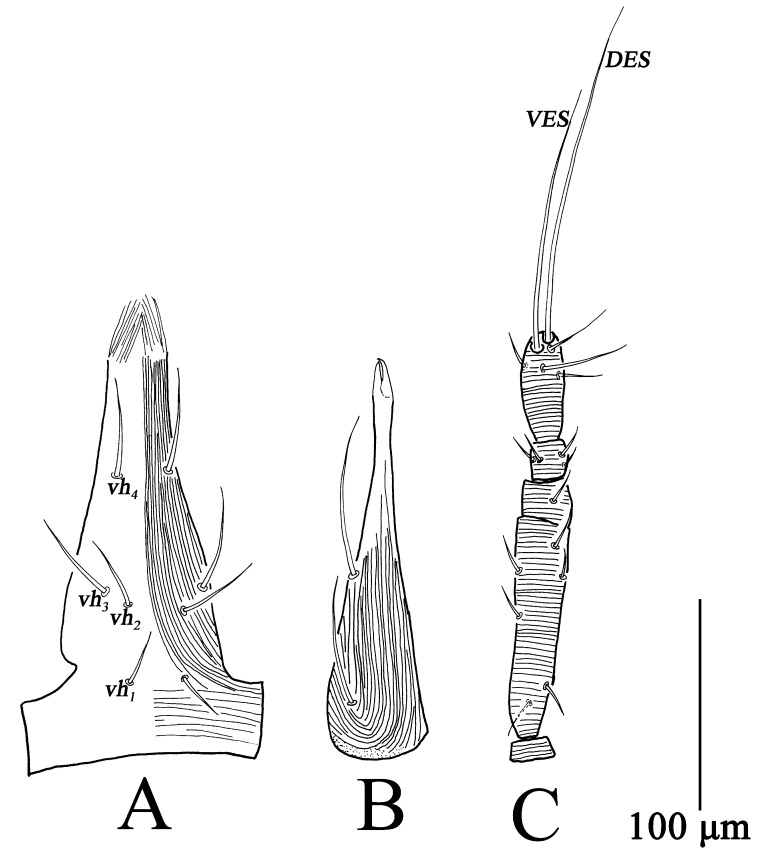
*Bdella muscorum*, deutonymph: (**A**) Subcapitulum; (**B**) Chelicera; (**C**) Palp.

**Figure 8 insects-13-01080-f008:**
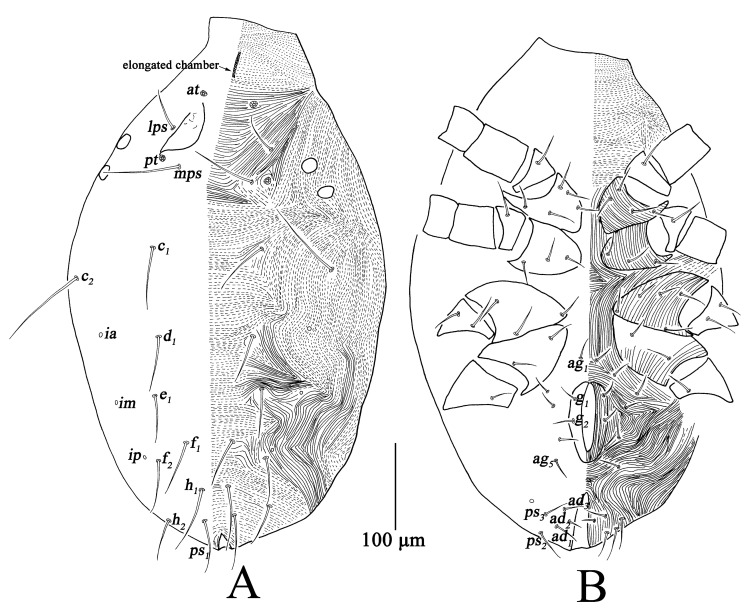
*Bdella muscorum*, deutonymph: (**A**) Dorsal view of idiosoma; (**B**) Ventral view of idiosoma.

**Figure 9 insects-13-01080-f009:**
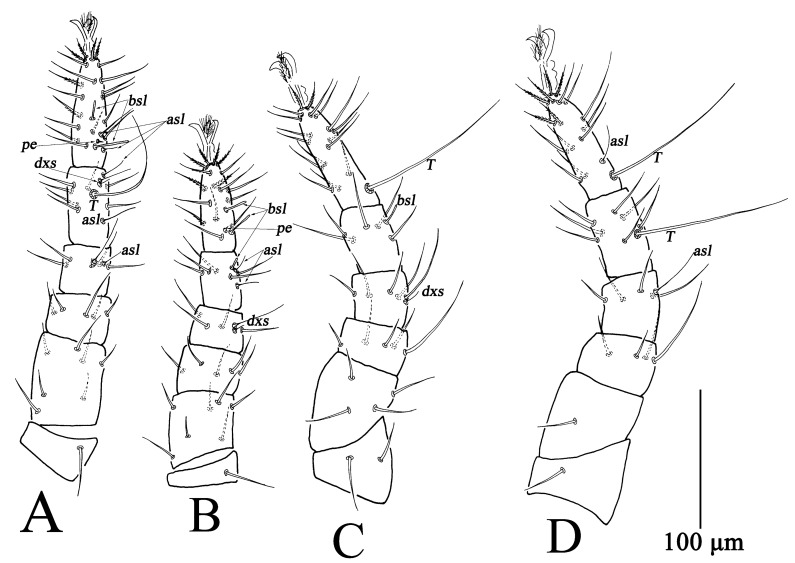
*Bdella muscorum*, deutonymph: (**A**) Leg I; (**B**) Leg II; (**C**) Leg III; (**D**) Leg IV.

**Figure 10 insects-13-01080-f010:**
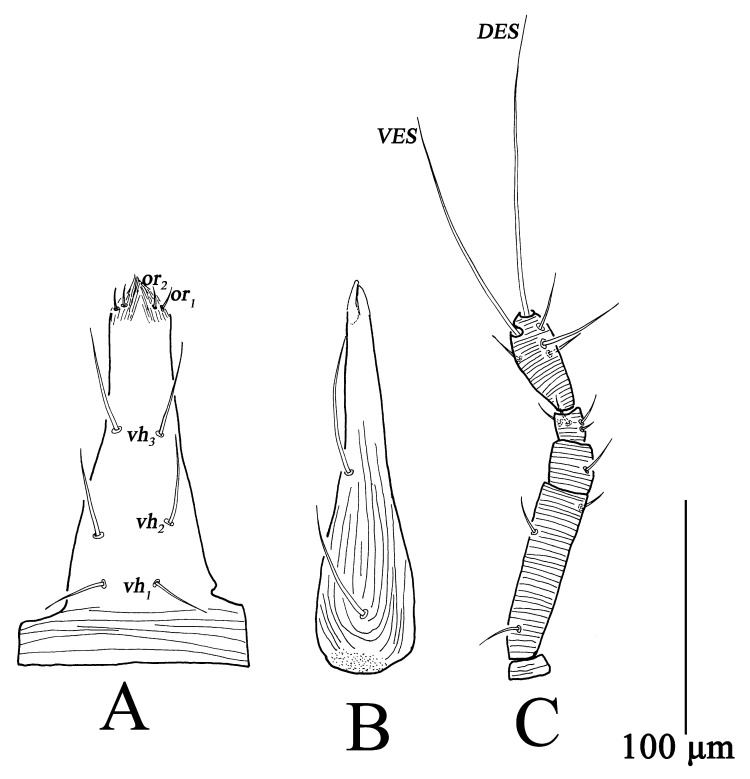
*Bdella muscorum*, protonymph: (**A**) Subcapitulum; (**B**) Chelicera; (**C**) Palp.

**Figure 11 insects-13-01080-f011:**
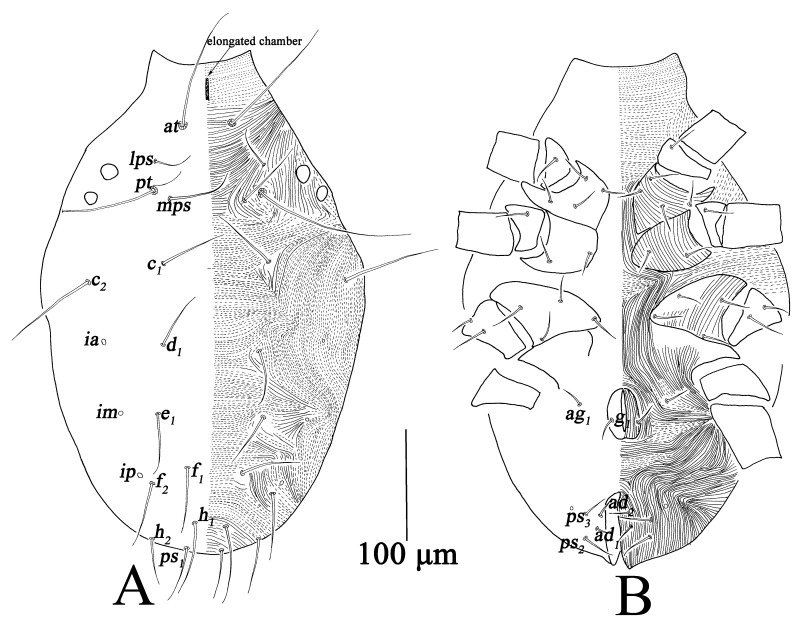
*Bdella muscorum*, protonymph: (**A**) Dorsal view of idiosoma; (**B**) Ventral view of idiosoma.

**Figure 12 insects-13-01080-f012:**
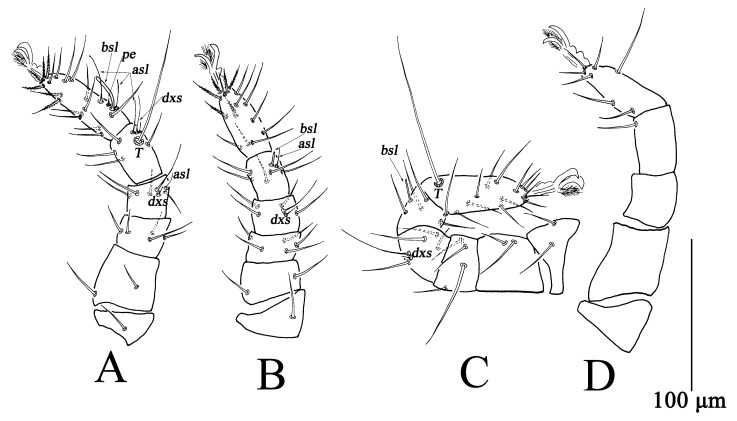
*Bdella muscorum*, protonymph: (**A**) Leg I; (**B**) Leg II; (**C**) Leg III; (**D**) Leg IV.

**Figure 13 insects-13-01080-f013:**
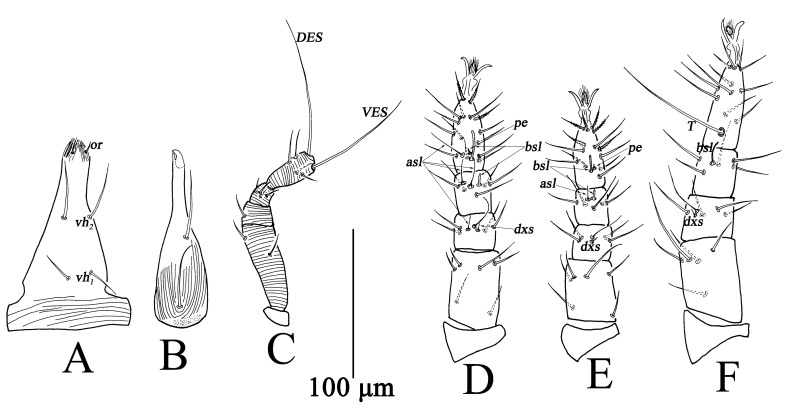
*Bdella muscorum*, larva: (**A**) Subcapitulum; (**B**) Chelicera; (**C**) Palp; (**D**) Leg I; (**E**) Leg II; (**F**) Leg III.

**Figure 14 insects-13-01080-f014:**
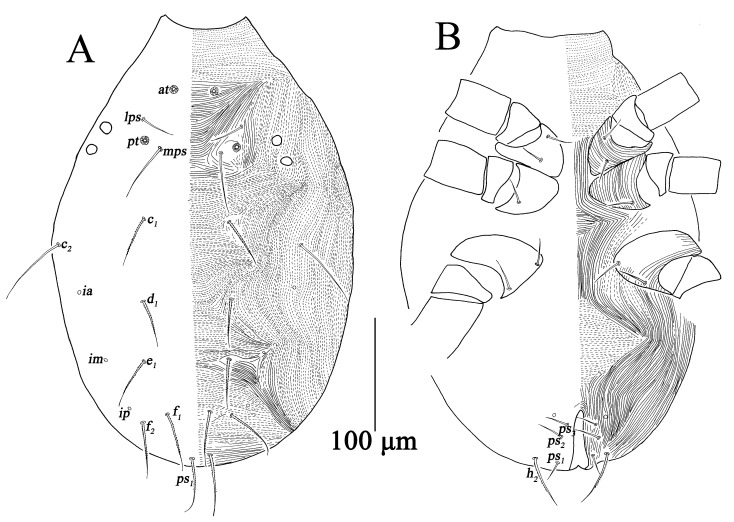
*Bdella muscorum*, larva: (**A**) Dorsal view of idiosoma; (**B**) Ventral view of idiosoma.

**Table 1 insects-13-01080-t001:** Changes of chaetotaxy in different stages of *Bdella muscorum*.

Stage	Dorsal Hypostomal Setae (*vh*)	Basifemur Setae of Palp	Aggenital Setae (*ag*)	Genital Setae (*g*)	Anal Setae (*ad*)	Postanals (*ps*)	Legs
Adult (Female)	6	9–11	11	8	3	3	4
Tritonymph	5	7	9	5	3	3	4
Deutonymph	4	6	5/6	2	3	3	4
Protonymph	3	3	1	1	2	3	4
Larva	2	2	0	0	0	3	3

**Table 2 insects-13-01080-t002:** Changes of leg chaetotaxy in different stages of *Bdella muscorum*.

Stage	Leg	Coxa	Trochanter	Basifemur	Telofemur	Genu	Tibia	Tarsus
Adult (Female)	I	6	1	12(13)	9(10)	6 *sts*, 2 *asl*, 1 *dxs*	14 *sts*, 3 *asl*, 1 *dxs*, 1 *T*	27 *sts*, 2 *asl*, 2 *bsl*, 1 *pe*
	II	6	1	9(8)	9(8)	6 *sts*, 1 *dxs*	11 *sts*, 2 *asl*, 1 *bsl*	23 *sts*, 2 *bsl*, 1 *pe*
	III	5	2	11(9)	5(6)	6 *sts*, 1 *dxs*	12 *sts*, 1 *asl*	27 *sts*, 1 *T*
	IV	3	2	5	8	8 *sts*, 1 *asl*	13 *sts*, 1 *T*	23 *sts*, 1 *asl*, 1 *T*
Tritonymph	I	5	1	8	6	5 *sts*, 2 *asl*, 1 *dxs*	8 *sts*, 2 *asl*, 1 *dxs*, 1 *T*	26 *sts*, 2 *asl*, 2 *bsl*, 1 *pe*
	II	5	1	7	5	5 *sts*, 1 *dxs*	7 *sts*, 2 *asl*, 1 *bsl*	21 *sts*, 2 *bsl*, 1 *pe*
	III	5	2	7	6	4 *sts*, 1 *dxs*	7 *sts*, 1 *asl*	21 *sts*, 1 *T*
	IV	3	1	3	5	6 *sts*, 1 *asl*	8 *sts*, 1 *T*	19 *sts*, 1 *asl*, 1 *T*
Deutonymph	I	5/4	1	7	5	4 *sts*, 1 *asl*, 1 *dxs*	7 *sts*, 2 *asl*, 1 *dxs*, 1 *T*	19 *sts*, 2 *asl*, 2 *bsl*, 1 *pe*
	II	3	1	7	5	4 *sts*, 1 *dxs*	5 *sts*, 2 *asl*, 1 *bsl*	16 *sts*, 2 *bsl*, 1 *pe*
	III	3/5	2	4	5	4 *sts*, 1 *dxs*	5 *sts*, 1 *bsl*	17 *sts*, 1 *T*
	IV	2	1	1	4	4 *sts*, 1 *asl*	7 *sts*, 1 *T*	15 *sts*, 1 *asl*, 1 *T*
Protonymph	I	4	1	2	5	4 *sts*, 1 *asl*, 1 *dxs*	4 *sts*, 1 *asl*, 1 *dxs*, 1 *T*	17 *sts*, 2 *asl*, 1 *bsl*, 1 *pe*
	II	2	1	2	5	4 *sts*, 1 *dxs*	5 *sts*, 1 *asl*, 1 *bsl*	15 *sts*
	III	4	2	2	4	4 *sts*, 1 *dxs*	5 *sts*, 1 *bsl*	13 *sts*, 1 *T*
	IV	0	0	0	0	0	1	7 sts
Larva	I	2	0	6		4 *sts*, 1 *asl*, 1 *dxs*	5 *sts*, 2 *asl*	14 *sts*, 1 *asl*, 1 *bsl*, 1 *pe*
	II	1	0	6		4 *sts*, 1 *dxs*	5 *sts*, 1 *asl*, 1 *bsl*	13 *sts*, 1 *bsl*, 1 *pe*
	III	2	0	5		4 *sts*, 1 *dxs*	5 *sts*, 1 *bsl*	11 *sts*, 1 *T*

## Data Availability

Data is available on request.

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
