# Peer review of "Redescription of Bdella muscorum Ewing, 1909 (Bdellidae: Bdellinae) from China with Its First Description of Ontogenyâ€"

_insects, 2022, doi:10.3390/insects13121080_

Round 1
Reviewer 1 Report
Comments
Redescription of Bdella muscorum Ewing, 1909 (Bdellidae: Bdellinae) from China with its first description of ontogeny
by
Youfang Wu, Khan Samiullah, Daochao Jin, Tianci Yi and Jianjun Guo
The submitted paper contains a redescription of Bdella muscorum Ewing, 1909 (Bdellidae: Bdellinae) from China. Bdellidae are active predators of small arthropods that have been shown to be effective biological control agents against spider mites and springtails: Hypogastrura communis and Sminthurus viridis. In this study, Bdella muscorum is redescribed based on the adult. Besides, all immature stages are described for the first time. An original key to the known Chinese Bdella species is provided as well.
The paper is written in a standard array, the tables are informative and the figures are of proper quality. The reviewed paper represents a typical alpha-taxonomic kind. Some corrections are marked in the enclosed text. Nevertheless, materials and methods are not described adequately. There is no data about the rearing process of immature stages. It is the only way to identify and correlate all life stages of the species! How do the authors know that the individuals representing all life stages collected during the study do in fact belong to the same species? The rearing under controlled conditions is the fundamental process that should be explained and described in the chapter “Material and Methods”.
To conclude, I am of an opinion that the article fits into the scope of “Insects” and could be published, but after a major revision according to remarks in the enclosed text has been done.

Author Response
Point 1: Not in all taxa, e.g. Trombiculidae. I propose "Many species of mites"
Response 1: We have revised in the manuscript (Line 12). And other inappropriate word and Error writing were corrected according to reviewer’s comments. All corrected were tried to identify by highlighting color mark in the manuscript.
Point 2: There is no data about rearing process. It is the only way to synonymize all life stages of the species! How the authors know, that obtained material, embracing all life stages, belong to the same species? Rearing in captivity is the fundamental process, that should be explained and described in this chapter.
Response 2: All life stages of the species that we were collected in the same habitat and the same place. We know that they are the same species based on distinguishing characteristics of the species.
Reviewer 2 Report
Dziękujemy za zaproszenie na spotkanie z Twoim manuskryptem zatytułowanym: Redescription of Bdella muscorum Ewing, 1909 (Bdellidae: Bdellinae) z Chin z pierwszym opisem ontogenezy.
Badanie jest interesujÄ…ce i wypeÅ‚nia lukÄ™ w tej dziedzinie wiedzy. RÄ™kopis jest na ogóÅ‚ dobrze napisany, zwięźle i zwięźle. Wyniki sÄ… jasno przedstawione. Odpowiednio wykorzystano materiaÅ‚ i metody. ProszÄ™ rozważyć poprawki:
Linia 45-46: "Próbki zostaÅ‚y wyekstrahowane z próbek za pomocÄ… lejków Berlese-Tullgren, zamontowanych na szkieÅ‚kach w medium Hoyera [13],..."
ProszÄ™ uzupeÅ‚nić informacjÄ™: ile dni prób zostaÅ‚o wyodrÄ™bnionych
interpunkcja:
Wiersz 97: "Bdella muscorum" wszystkie czcionki kursywÄ…
Linia 315: przecinek nie jest potrzebny
Linia 322: usuń spację
Proponuję pogrubioną czcionkę dla wszystkich oznaczeń liter na rysunkach.
Author Response
Point 1: Linia 45-46: "Próbki zostaÅ‚y wyekstrahowane z próbek za pomocÄ… lejków Berlese-Tullgren, zamontowanych na szkieÅ‚kach w medium Hoyera [13],..."
ProszÄ™ uzupeÅ‚nić informacjÄ™: ile dni prób zostaÅ‚o wyodrÄ™bnionych
Response 1: Specimens were extracted from the samples by using Berlese-Tullgren funnels for 12–24 hours.
Point 2: interpunkcja:
Wiersz 97: "Bdella muscorum" wszystkie czcionki kursywÄ…
Linia 315: przecinek nie jest potrzebny
Linia 322: usuń spację
Proponuję pogrubioną czcionkę dla wszystkich oznaczeń liter na rysunkach.
Response 2: We have corrected in the manuscript (Line 97, 315, 322). And all letter marks in the figures have been bolded.
Reviewer 3 Report
1. There is something wrong with the English grammars in the manuscript, and it should be modified, especially by a native speaker.
2. The detailed modification suggestions can be found in the manuscript, please check them.
3. In the section of “Voucher material”, please paired the latitude and longitude with altitude correspondently.
4. “According to the description of B. muscorum from different country, we found the number of leg setae are different in different area, which is relatively constant” The sentence makes me confused, and check it please. Maybe it should be:According to the description of B. muscorum from different country, we found the number of leg setae are different in different area, which is relatively inconstant”.

Author Response
Point 1: There is something wrong with the English grammars in the manuscript, and it should be modified, especially by a native speaker.
Response 1: We have revised the English grammars according to reviewer’s comments. And poor sentences were tried to reconstruct, inappropriate word choice were done, plural and tense errors throughout the manuscript were corrected.
Point 2: The detailed modification suggestions can be found in the manuscript, please check them.
Response 2: We have modified according to reviewer’s comments. And all modification were highlighted in the manuscript.
Point 3: In the section of “Voucher material”, please paired the latitude and longitude with altitude correspondently.
Response 3: We have revised and paired the latitude and longitude with altitude correspondently in section of “Voucher material”.
Point 4: “According to the description of B. muscorum from different country, we found the number of leg setae are different in different area, which is relatively constant” The sentence makes me confused, and check it please. Maybe it should be:According to the description of B. muscorum from different country, we found the number of leg setae are different in different area, which is relatively inconstant”.
Response 4: We have rewritten the sentence in the manuscript.
Round 2
Reviewer 1 Report
This sentence should be included in "Material and methods" chapter:
All life stages of the species that we were collected in the same habitat and the same place. We know that they are the same species based on distinguishing characteristics of the species.
Author Response
Point 1: This sentence should be included in "Material and methods" chapter:
All life stages of the species that we were collected in the same habitat and the same place. We know that they are the same species based on distinguishing characteristics of the species.
Response 1: We have added the sentence “All life stages of the species that we were collected in the same habitat and the same place. We know that they are the same species based on distinguishing characteristics of the species” in section of “Material and methods".